# Hydrogen Delocalization in an Asymmetric Biomolecule: The Curious Case of Alpha-Fenchol

**DOI:** 10.3390/molecules27010101

**Published:** 2021-12-24

**Authors:** Robert Medel, Johann R. Springborn, Deborah L. Crittenden, Martin A. Suhm

**Affiliations:** 1Institut für Physikalische Chemie, Georg-August-Universität Göttingen, Tammannstr. 6, 37077 Goettingen, Germany; j.springborn@stud.uni-goettingen.de (J.R.S.); msuhm@gwdg.de (M.A.S.); 2School of Physical and Chemical Sciences, University of Canterbury, Private Bag 4800, Christchurch 8140, New Zealand; deborah.crittenden@canterbury.ac.nz

**Keywords:** delocalization, tunneling, terpene, alcohol, vibrational spectroscopy, supersonic jet expansion, structural determination

## Abstract

Rotational microwave jet spectroscopy studies of the monoterpenol α-fenchol have so far failed to identify its second most stable torsional conformer, despite computational predictions that it is only very slightly higher in energy than the global minimum. Vibrational FTIR and Raman jet spectroscopy investigations reveal unusually complex OH and OD stretching spectra compared to other alcohols. Via modeling of the torsional states, observed spectral splittings are explained by delocalization of the hydroxy hydrogen atom through quantum tunneling between the two non-equivalent but accidentally near-degenerate conformers separated by a low and narrow barrier. The energy differences between the torsional states are determined to be only 16(1) and 7(1) cm${}^{-1}hc$ for the protiated and deuterated alcohol, respectively, which further shrink to 9(1) and 3(1) cm${}^{-1}hc$ upon OH or OD stretch excitation. Comparisons are made with the more strongly asymmetric monoterpenols borneol and isopinocampheol as well as with the symmetric, rapidly tunneling propargyl alcohol. In addition, the third—in contrast localized—torsional conformer and the most stable dimer are assigned for α-fenchol, as well as the two most stable dimers for propargyl alcohol.

## 1. Introduction

In chemistry, it is well known that quantum tunneling of particles between potential wells separated by finite barriers enhances reaction rates [1], but it is less widely recognised that it also can result in splitting of energy levels and delocalization [2]. The latter consequences are experimentally well characterized for coherent tunneling between equivalent wells in many small- to medium-sized systems in the gas phase or/and at cryogenic temperatures, e.g., in jet expansions. In contrast, quantum tunneling effects are seldom observed within biomolecular systems that typically feature larger molecular structures or/and are affected by environmental perturbations, resulting in an absence of symmetry. However, localization with increasing asymmetry is a gradual, not abrupt, process. As a rule of thumb, the particle remains strongly delocalized if the introduced energetic asymmetry is not larger than the tunneling splitting in the symmetric case [3,4,5]. The description of tunneling in such slightly asymmetric double-minimum potentials continues to draw theoretical interest [6,7,8,9,10,11,12,13,14].

Known experimental splittings for hydrogen atom tunneling span a range of more than eight orders of magnitudes, with most of them being (far) smaller than 1 cm${}^{-1}hc$ ($\hat{\approx }$0.01 kJ mol${}^{-1}$) [15]. For this reason, localization is the typical, but not the strict, outcome from symmetry breaking by chemical [3,16,17,18,19] or isotopic [4,5,20,21,22,23,24,25,26,27,28] substitution, internal rotation of molecular groups [6,16,20,24,29], or from environmental influences such as solvation [16,30], matrix embedding [3,31,32] or crystallization [33]. However, delocalization can still be observed upon excitation if either the barrier [3,5,21,25,26,29] or the energetic asymmetry [22,34] are sufficiently lowered.

There are only a few asymmetric systems known to exhibit substantial atomic delocalization in their vibrational and electronic ground states. Among them are certain isotopologs of the well-studied dimer of hydrogen chloride. Tunneling splittings from geared internal rotation of (H${}^{35}$Cl)${}_{2}$ and (H${}^{37}$Cl)${}_{2}$ are as large as 15.5 and 15.4 cm${}^{-1}hc$, respectively [35]. The zero-point asymmetry between hypothetical localized H${}^{35}$Cl⋯H${}^{37}$Cl and H${}^{37}$Cl⋯H${}^{35}$Cl isomers, differing in the acceptor⋯donor roles for the hydrogen bond, is calculated to be on the order of 1–2 cm${}^{-1}hc$. This is an order of magnitude smaller than the tunneling splitting, indicating that this isotopically mixed dimer is highly delocalized. Still, the symmetry-breaking is sufficient to transfer spectral activity to additional transitions, which enables direct determination of the splittings in the ground and the HCl stretch excited states. In the latter, the splitting is reduced to about 3.2 cm${}^{-1}hc$ for the symmetric dimers. Increasing tunneling mass by double-deuteration reduces the tunneling splittings to 6.0 and 1.2 cm${}^{-1}hc$ in the ground and DCl stretch excited states, respectively [36]. The latter is smaller than the asymmetry of 3.0 cm${}^{-1}hc$ between the DCl stretch excited mixed D${}^{35}$Cl⋯D${}^{37}$Cl and D${}^{37}$Cl⋯D${}^{35}$Cl dimers, which are therefore better described as distinguishable localized species, in contrast to both the vibrational ground state and the parent isotopolog.

Symmetric parents for other well investigated structural families with respect to (partial) localization by chemical or isotopic symmetry breaking include porphycen (tunneling splitting of 4.4 cm${}^{-1}hc$) [37,38,39,40,41], malonaldehyde (21.6 cm${}^{-1}hc$) [4,7,14] and 9-hydroxyphenalenone (69 cm${}^{-1}hc$) [3,42,43,44,45].

Considering the vast structural diversity found in nature, it is perhaps not surprising, at least in hindsight, that tunneling splittings and partial delocalization also occur in inherently asymmetric biomolecules, i.e., systems which cannot be derived from a symmetric parent by minor modification. In this article, we present spectroscopic and quantum-chemical evidence for this phenomenon in the monoterpenol α-fenchol (Figure 1).

This compound and its oxidized form fenchone are common natural products produced by many plants [46], including fennel (German: Fenchel), from which they derive their names [47]. α-Fenchol is used as a fragrance in cosmetic products, household cleaners and detergents [48].

The isomerism of α-fenchol was previously investigated by Pate et al. in a neon expansion with microwave spectroscopy [49]. The three expected torsional conformers differ in the orientation of the light hydroxy hydrogen atom relative to the heavy and rigid molecular frame (Figure 2). A single dominant conformer was observed which was assigned to g− based on the good agreement between experimental and computed values for rotational constants and relative electric dipole moment components, as well as on computational prediction that g− is the global minimum energy conformer, 0.3 kJ mol${}^{-1}$ below g+ at B3LYP-D3(BJ)/6-311++G(d,p) level. The calculated properties of the other conformers were not discussed, as the study focused on the determination of the enantiomeric excess via chiral tag spectroscopy.

A more definitive investigation of the diastereomerism of fenchol (including the epimer β-fenchol with the relative configuration at the alpha carbon atom inverted (Figure 1)) was done in parallel by Neeman and Huet, using the same experimental approach [50]. In contrast to Pate et al., they assessed that a conformational assignment for α-fenchol on the basis of calculated rotational constants, dipole moment components and relative energy is not possible, as these are too similar between g− and g+. Due to the challenges faced, they termed the conformational landscape a “maze” and nicely illustrated it as such in the table of content figure. Looking for further clues, they pointed out that the asymmetry parameter for the rotational constants agrees better with the prediction for the g− conformer, a tendency which is further supported by the substitution geometry and the nuclear quadrupole hyperfine structure obtained after deuteration. Thus, they eventually reached the same assignment as Pate et al. Still, the absence of the second most stable g+ conformer, despite being predicted to be almost degenerate in energy (down to 0.03 kJ mol${}^{-1}$ at MP2/6-311++G(d,p) level), remained somewhat puzzling. It was postulated that it may not be confined to its torsional potential well due to a very low computationally-predicted separating barrier that may be overcome by zero-point vibrational motion.

Recently, we were able to assign all three expected conformers of both borneol and isopinocampheol [51], two constitutional isomers of α-fenchol (Figure 1), by taking advantage of the differences in their OH stretching fundamental wavenumbers using Raman jet spectroscopy. In the present work, we extend our monoterpenol investigation to FTIR jet spectroscopy as well as α-fenchol. The complexity of the obtained spectra for α-fenchol is in stark contrast to the other two monoterpenols which indicates that the conformational landscape maze [50] is even deeper than realized so far, but we also propose a non-classical way to escape it. As it turns out, allegorically speaking, there is not a single exit but one has to consider breaking through a barrier.

## 2. Materials and Methods

### 2.1. Experimental Techniques

Gas mixtures were prepared by enriching 1.6 bar of helium with the vapor pressure of the respective compound at a set temperature. (+)-α-Fenchol (25 ${}^{\circ }$C, 96.8%, Alfa Aesar), (−)-borneol (24 ${}^{\circ }$C, 98.9%, Acros Organics), (−)-isopinocampheol (24 ${}^{\circ }$C, 98%, Sigma Aldrich) and propargyl alcohol (−10 to +23 ${}^{\circ }$C, 99%, abcr GmbH) were used as supplied. (+)-α-Fenchol-OD was generated by dissolving the commercial sample in an excess of MeOD (99%, Eurisotop), leaving the solution for half an hour for isotope exchange, evaporating methanol under reduced pressure and drying for 30 min in vacuum.

In the Raman jet setup [52], the respective gas mixture was expanded continuously at room temperature from a pressure between 0.7–0.8 bar through a (4 × 0.15) mm${}^{2}$ slit nozzle into an evacuated chamber. The expansion was probed at different distances from the nozzle by a Spectra Physics Millenia eV laser (532 nm, cw, 20–24 W). The scattered light was collected perpendicular to the propagation directions of both laser and jet with a camera lens and focused onto a one meter monochromator (McPherson). Photons from Stokes Raman scattering were co-added by a LN${}_{2}$-cooled CCD-camera (Princeton, PyLoN 400) over several minutes and averaged over multiple repetitions. The combination of laser and monochromator results in a spectral resolution of about 1 cm${}^{-1}$. The spectra were calibrated with neon vacuum transitions, we assume spectral positions and their differences to be accurate up to ±1 cm${}^{-1}$.

In the FTIR jet setup [53], the gas mixture was expanded at a pressure between 0.10–1.20 bar through a (600 × 0.2) mm${}^{2}$ slit nozzle into vacuum. The pulsed supersonic expansion was probed by a synchronized FTIR scan at 2 cm${}^{-1}$ resolution with a Bruker IFS 66 v/S spectrometer (single-sided-fast-return mode), with averaging over typically several hundred repetitions. For the recording of stationary vapor spectra at ambient temperature, the same compound/helium gas mixtures (at a total pressure of 0.40 bar) and the same spectral parameters were used, averaged over one minute of background and one minute of sample scans. To correct for obstructing signals from drifting concentration of residual water vapor in the optical path (reduced by evacuation or flushing with dried air), a spectrum without sample was scaled to characteristic water lines in the 3500–4000 cm${}^{-1}$ region of each FTIR spectrum and subtracted.

### 2.2. Computational Techniques

Density functional theory (DFT) and second-order Møller–Plesset perturbation theory (MP2) computations were carried out with the Gaussian 09 Rev. E.01 program package [54]. The B3LYP [55,56,57,58], PBE0 [59], and B2PLYP [60] functionals together with Grimme’s D3 dispersion correction without three-body term but with Becke–Johnson damping [61] were used in conjunction with the minimally augmented may-cc-pVTZ basis set [62] and an ultrafine integration grid throughout. MP2 was combined with the 6-311++G(d,p) basis set and tight convergence for the self consistent field. For geometry optimizations, very tight convergence criteria were used with a subsequent frequency calculation in the double-harmonic approximation with Raman activities. Minima were confirmed by the absence of imaginary frequencies, and transition states by the presence of exactly one imaginary frequency. Single-point energies at the DLPNO-CCSD(T) level [63,64,65] with the aug-cc-pVQZ basis set were calculated at the DFT- or MP2-optimized geometries using ORCA version 4.2.1 [66]. Relaxed scans of the electronic energy were performed at a B3LYP-D3(BJ)/may-cc-pVTZ level with 1${}^{\circ }$ step size along the torsional dihedral coordinates τ(HCOH) for methanol and secondary alcohols, and τ(CCOH) for primary and tertiary alcohols. For symmetric alcohols, only the non-redundant part of the potential (e.g., 180${}^{\circ }$ for propargyl alcohol and 60${}^{\circ }$ for methanol) was scanned and then duplicated to preclude symmetry reduction by numerical noise. Specifics sets of commands and keywords are available in the electronic Appendix A (ESI).

To simulate spectra for localized conformers, calculated Raman activities and depolarization ratios were converted to scattering cross sections, accounting for the laser wavelength and polarization-dependent sensitivity of monochromator and camera [67,68]. This is detailed in the ESI. Calculated infrared band strengths were used directly. Relative populations are based on the assumption that the conformational cooling stops at about 100 K and that vibrational and rotational partition functions for the conformers are similar enough to justify the use of zero-point corrected relative energies instead of Gibbs energies. The value of 100 K corresponds to an estimate at a detection distance from the nozzle of 1 mm [69], using the accurately known conformational energy difference of ethanol [70].

Numerical solutions to the 1D torsional Schrödinger equation were obtained using the Chung–Phillips algorithm [71], employing reduced moments of inertia that account for minor counter-rotation of the alcohol scaffold upon hydroxy group rotation. The code is available on github, at https://github.com/dlc62/Torsion1D (accessed on 18 December 2021). To account for the dependence of the effective OH bond length on the OH stretching quantum number as well as on isotopic substitution, the calculated equilibrium value is increased based on experimental data for the rotational constant of the hydroxyl radical [72,73]. This is detailed in Appendix A in the ESI.

## 3. Results and Discussion

### 3.1. Spectra of Monomeric α-Fenchol

We have previously explored FTIR jet spectra of α-fenchol in the context of chirality recognition with α-pinene [74]. In the high-wavenumber part of the OH stretching region, they show two bands at 3675 and 3666 cm${}^{-1}$, with the latter having a higher intensity (Figure 3).

This is in apparent agreement with the prediction of at least two of the conformers (g− and g+) being close in energy and separated by 8–9 cm${}^{-1}$ in their harmonic OH stretching wavenumbers (Table 1).

However, more detailed analysis reveals that this match is superficial. The g− conformer is almost consistently predicted to be the lowest in energy (in agreement with the conclusion from microwave spectroscopy [49,50]) and to feature the highest OH stretching wavenumber, which is in conflict with the observed sequence. The IR band strengths of these two conformers are predicted to be almost equal at all employed levels of theory and therefore cannot explain this discrepancy, either. Furthermore, the absolute band positions are not in good agreement with the predictions from a recently proposed model based on harmonic PBE0 calculations [51]. The model captures the OH stretching wavenumbers of 46 alcohols conformers within ±3 cm${}^{-1}$ including those of borneol and isopinocampheol. For α-fenchol, however, the errors would reach about 5 cm${}^{-1}$ for both conformers (simulation in the upper part of Figure 3) and would further escalate if the assignment was reversed. Finally, the FTIR spectrum of the vapor at ambient temperature has a spectral center of gravity substantially shifted to lower wavenumbers relative to the jet. This can only partially be explained by the expected presence of the *t* conformer at a lower wavenumber, as it has a predicted high relative energy at most levels of theory (Table 1) and, more importantly, an IR band strength only about half as large. This is confirmed by comparison with the FTIR spectra of borneol (Appendix A) and isopinocampheol (Appendix A) which feature *t* conformers with similar calculated properties (Appendix A).

The Raman jet spectra (lower half of Figure 3) help unravel this mystery by revealing five additional bands in this spectral region. This is thanks to superior resolution and sensitivity of the Raman jet detection for alcohol monomer investigations, as well as higher conformational temperatures due to exclusive detection closer to the nozzle enabled by the narrow laser beam. One of these bands, at 3657 cm${}^{-1}$, reflects a downside of the method for this purpose, though, as it is caused by the mostly Raman-active symmetric stretching mode of a common water impurity which might obstruct alcohol signals. The two signals from the FTIR jet spectra are confirmed and retain their relative intensities upon variation of the detection distance. Three newly detected bands at 3659, 3650 and 3646 cm${}^{-1}$ in contrast lose roughly half of their relative intensity at 2.0 mm distance compared to 0.5 mm, while a very weak band at 3633 cm${}^{-1}$ instead gains relative intensity.

Thus, how to explain the presence of six bands for α-fenchol, when only a single band for each of the three conformers was observed for borneol and isopinocampheol [51]? One possibility to consider are other impurities beyond water. Both microwave investigations reported β-fenchol as a minor component in the commercial samples of α-fenchol with spectral intensities two orders of magnitude smaller [49,50]. The purity stated by the manufacturer for the sample used in the present work is 96.8%. The harmonic OH stretching wavenumbers for the analogous conformers of α- and β-fenchol are predicted to be within 2 cm${}^{-1}$ or less (Appendix A). Any possible spectral contribution of β-fenchol would be too minor and too similar to be detectable at the available signal-to-noise ratio and resolution and thus cannot explain the additional signals (Appendix A).

Not technically impurities, but other species to consider, are dimers of α-fenchol. They are expected to feature a dangling hydroxy group for the acceptor molecule of the hydrogen bond, typically scattering slightly downshifted from the monomer signals, as it is known from other saturated alcohols [75,76,77]. As aggregation builds up in the colder part of the expansion more distant from the nozzle, this is a plausible assignment for the weak signal at 3632 cm${}^{-1}$ gaining intensity. This assignment as well as the strongly downshifted donor mode will be discussed further below in Section 3.8. No dimer signals (neither acceptor nor donor) could be observed for borneol and isopinocampheol, as their vapor pressures are an order of magnitude smaller compared to α-fenchol.

This leaves five signals for assignment. One hint comes from a pattern of repeating spectral differences between four of these bands, highlighted in the center of Figure 4. A possible interpretation is the four energy level scheme shown in the same figure, in which spectral separations are translated into ground and excited state splittings with the method of combination differences. Such an assignment also has the beauty of explaining the observed changes in relative intensities. If the two high-wavenumber signals at 3675 and 3666 cm${}^{-1}$ do not originate from different conformers, but from the same state, it becomes understandable why their relative intensities are unaffected by changes in the experimental conditions. Likewise, the two signals at 3659 and 3650 cm${}^{-1}$ lose relative intensity at the same rate under colder conditions if they arise from the same excited state.

A possible mechanism to explain the presence of additional bands in the spectrum of α-fenchol, and their absence in those of borneol and isopinocampheol, is a tunneling interaction between the two *g* conformers in the case of a near-degeneracy, leading to delocalization of the hydroxy hydrogen atom in the OH stretching ground and/or the excited state. Similar spectral patterns have been assigned previously for tunneling molecules, namely for ethanol tentatively in the OH stretching fundamental and first overtone regions [78], as well as for multiple fundamental excitations of malonaldehyde [32], (HCl)${}_{2}$ [35,36] and (HBr)${}_{2}$ [79]. A tunneling interaction would also increase the energy difference between the torsional states but not shift the centers of gravity for states and transitions, according to a first-order quantum coupling model. Indeed, the wavenumber average of these four bands (3662.5 cm${}^{-1}$) is in better agreement with the prediction for the average of the localized *g* conformers according to the model based on PBE0 (3665.2 cm${}^{-1}$) [51]. However, the required accidental energy near-degeneracy between non-equivalent conformations in their vibrational ground and/or OH stretch excited states is far from a routine observation, in particular for non-isotopologic symmetry breaking. The plausibility of this hypothesis will be investigated computationally in more detail in Section 3.3. The final band at 3646 cm${}^{-1}$ can be assigned to the high-energetic and therefore localized *t* conformer, in excellent agreement with its predicted position of 3645.8 cm${}^{-1}$.

In an attempt to localize the hydrogen atom also in the *g* potential wells, and therefore simplify the spectrum, we deuterated the hydroxy group. While deuteration does indeed have a substantial impact on the observed spectral pattern, the changes are not as anticipated. Instead of collapsing into into the classically expected transitions, transition intensities become more evenly distributed between the same number of bands.

Broad bands at 2733 and 2723 cm${}^{-1}$ in the OD stretching region of the Raman spectrum (Figure 5) are already present before deuteration and can likely be attributed to some overtone or combination excitations not involving the hydroxy hydrogen.

Other spectral activity is readily assigned to impurities: HDO [80] (2723 cm${}^{-1}$, coinciding with the scaffold overtone/combination band), D${}_{2}$O [80,81] (2671 cm${}^{-1}$) and residues of the used deuteration reagent MeOD [82] (2718 cm${}^{-1}$).

An eye-catching feature of this spectrum, again due to repeating separations, is a band pattern consisting of four components with similar intensities at 2707, 2704, 2700 and 2697 cm${}^{-1}$. In analogy to the protiated alcohol, a possible interpretation is an assignment to a four-level scheme, as shown in Figure 6 with torsional splittings of 7 and 3 cm${}^{-1}hc$, reduced from 16 and 9 cm${}^{-1}hc$ by deuteration. The spectral center of gravity of 2702 cm${}^{-1}$ is slightly smaller than the predicted [51] average of 2705.8 cm${}^{-1}$ for the localized *g* conformers, as for the protiated alcohol.

The relative intensities within this band pattern change only very slightly upon increasing the detection distance from 0.5 to 2.0 mm, which is consistent with a smaller torsional splitting of the OD stretch ground state as a reduced driving force for relaxation.

The FTIR jet spectrum (Figure 5), detected at a larger average distance from the nozzle, shows clear absorption for the unresolved 2707 and 2704 cm${}^{-1}$ transitions, while the 2700 and 2697 cm${}^{-1}$ signals only barely exceed noise level. This intensity discrepancy might be explained by differences in population or/and IR band strength; the more even absorption of the vapor at ambient temperature suggests the former influence to be the dominant one. We therefore conclude that the splitting of 7 cm${}^{-1}hc$ applies to the OD stretch ground-state and the one of 3 cm${}^{-1}hc$ to the excited state. This assignment is also supported by theoretical considerations, as will be elaborated in Section 3.3.

Strong parallels between this interpretation and the one for the hydrogen chloride dimer, summarized in the introduction, are apparent. This includes values of absolute splittings, their changes with both deuteration and stretching excitation, as well as the observation of additional transitions for slightly asymmetric species.

A less intense band at 2689 cm${}^{-1}$ gains relative intensity closer to the nozzle and its position is in excellent agreement with the model prediction (2689.5 cm${}^{-1}$) for the *t* conformer. Based on the opposite distance dependence of its relative intensity, the 2680 cm${}^{-1}$ signal is assigned to the acceptor of the most stable isomer of the deuterated dimer of α-fenchol. A weak signal at 2713 cm${}^{-1}$ is only detected very close to the nozzle. This and the up-shifted position might hint at an origin from an even higher torsional state, similar to the situation for ethanol [76,83] and methanol [83]. No counterpart is observed for α-fenchol-OH though.

An interesting question is: Why were transitions from the second conformer (or more accurately: torsional state) not observed in the microwave investigations [49,50], despite the high detection sensitivity of this technique and low energy differences of only 16 (OH) and 7 cm${}^{-1}hc$ (OD), equivalent to less than 0.2 and 0.1 kJ mol${}^{-1}$? For other tunneling systems, very efficient relaxation between the split states was concluded, approaching or reaching effective temperatures estimated in the single-digit Kelvin range [32,35,36,51,84]. Notably, no splitting from internal rotation of the hydroxy group was reported in a neon jet investigation of but-2-yn-1-ol [85], despite it featuring a calculated barrier of similar height to prop-2-yn-1-ol (propargyl alcohol), for which a second set of transitions was observed at ambient temperature [86,87] and in an argon jet [88]. We infer that, when using a strong relaxant agent [89] at a high pressure, such as neon at 2.0 or 4.5 bar in the former studies of α-fenchol, and a large average detection distance from the nozzle, as typical for microwave spectroscopy, there is the possibility that the depopulation of the upper tunneling state might become too extensive for it to be observed. This issue was also reported in the IR laser investigation of the HCl and DCl dimer and resolved by a switch from neon to argon and to a lower carrier gas pressure [35,36].

### 3.2. Consistency Checks for Isotope Effects

With simple models, it is possible to check whether the assignments made for protiated and deuterated α-fenchol are consistent with each other as well as with other alcohols.

Using a simple Morse oscillator model [90], the isotope effect on the ratio of the fundamentals can be expressed as:(1)$$\begin{array}{c} \frac{{\tilde{\nu }}_{i}}{{\tilde{\nu }}_{0}}=\frac{\sqrt{r}\omega_{e}-r2\omega_{e}x_{e}}{\omega_{e}-2\omega_{e}x_{e}} \end{array}$$
with ${\tilde{\nu }}_{i}$ and ${\tilde{\nu }}_{0}$ being the fundamental wavenumbers of the isotopically modified and the unmodified molecule, respectively, *r* the ratio of their reduced masses $\mu_{0}/\mu_{i}$, $\omega_{e}$ the harmonic wavenumber and $\omega_{e}x_{e}$ the anharmonicity constant of the unmodified molecule. If harmonic wavenumbers, mass ratios and anharmonicities do not differ too much, determined ratios of fundamental wavenumbers may be transferred between chemically related molecules. This is empirically known and used by spectroscopists for eight decades as the “ratio rule” [91]. With the OH stretching oscillator being very localized and decoupled (OD somewhat less so) and its anharmonicity being apparently widely unaffected by conformation and substitution [51], the requirements should be especially favorable for alcohols that are either not or only weakly hydrogen-bonded. This is indicated by observed uniform ratios for conformers of ethanol [76] and 1-propanol [77], and is here generalized across different alcohols. Using approximate but constant values of r=1/1.8868 [76] and $2\omega_{e}x_{e}=180\;{\mathrm{cm}}^{-1}$ (when also pragmatically including off-diagonal anharmonicity of about 10 ${\mathrm{cm}}^{-1}$) [51], the calculated fundamental ratios differ in the relevant $\omega_{e}$-range of about 3870–3800 cm${}^{-1}$ [51] only between 0.7377 and 0.7379. Replacing with the average value introduces an expected error of only ±0.3 cm${}^{-1}$ for this wavenumber range, provided that the assumptions are met.

The performance of this model is demonstrated in Figure 7 using a training set of available jet data [75,76,77,92] (Appendix A) spanning ranges of 53 (OH) and 40 cm${}^{-1}$ (OD) for 10 conformers of 5 alcohol monomers as well as acceptors of 6 alcohols dimers, which are only weakly perturbed by the hydrogen bond.

The correlation is captured by the proposed simple one-parameter scaling model with a mean absolute error of 0.4 cm${}^{-1}$ and a maximum absolute error of 1 cm${}^{-1}$, which is equal to the experimental uncertainty of the data points. The isotope effect on the spectral positions assigned to the *g* center of gravity, to the *t* conformer and to the dimer acceptor of α-fenchol fully meet the expectation, as evident from Figure 7.

Likewise, it has been shown that experimental tunneling splittings of protiated and deuterated alcohols (and also other series of compounds with hydrogen tunneling) can be directly correlated with each other as well, taking advantage of similar changes of the moment of inertia and the torsional wavenumber upon deuteration [93]. If the contributions to the assigned torsional splittings of α-fenchol are dominated by tunneling, and less by asymmetry, the isotope effect should be roughly captured by this model as well. Indeed, this is the case for the splittings of both the OH/OD stretch ground and excited states, as shown in Figure 8.

With the data points for α-fenchol being slightly above the regression line, this might indicate a larger impact of the asymmetry on the splitting of the deuterated isotopolog.

### 3.3. Theoretical Investigation of Torsional States of α-Fenchol

Unlike the systems mentioned in the Introduction, there is no obvious symmetric parent for α-fenchol to experimentally compare with. The closest open-chain analog is di-tert-butyl carbinol, which is however calculated to adopt as well an asymmetric conformation as a consequence of the steric repulsion between the bulky groups. Symmetric secondary alcohols with known tunneling splittings are 2-propanol [94], cyclohexanol [95] and cyclopropanol [96]. They feature experimental splittings at least an order of magnitude smaller, which correlate with calculated B3LYP-D3(BJ) barriers at least twice as high. In addition, the barrier of α-fenchol is also narrower, with the *g* equilibrium geometries calculated to be separated by Δτ(HCOH)= 104${}^{\circ }$, while the width of the barrier amounts to at least 129${}^{\circ }$ for the other mentioned secondary alcohols. In addition to the small difference in energy between the g+ and g− conformers, and the characteristic isotope effect on the splittings, this particularly low and narrow barrier for α-fenchol provides another piece of circumstantial evidence pointing towards the delocalization hypothesis.

To assess the torsional splitting, and the expected contribution from tunneling to it, we employ different computational models. As a first approach, it was demonstrated to be possible to reasonably estimate the tunneling splitting of symmetric alcohols only on the basis of calculated B3LYP properties of the enantiomeric *g* geometries and the transition state in between [93]. To model the limiting case of the potential of α-fenchol being perfectly symmetric, g−/g+ averages are used for the barrier heights and harmonic torsional wavenumbers in these models. Calculated splittings are given in Table 2 and are similar to the assigned ones (16 and 7 cm${}^{-1}hc$).

As an alternative, we solve the Schrödinger equation explicitly for a relaxed scan of the torsional potential. To validate the performance of the torsion 1D code in combination with the electronic B3LYP potential, we compare results with the experimental benchmarking data set for tunneling splittings for 27 symmetric alcohol species [20,87,94,95,96,97,98,99,100,101,102,103,104,105,106,107,108,109,110] compiled in Ref. [93]. The correlation is shown in Figure 9.

The experimental values are reproduced with a mean symmetric deviation factor MSDF (defined in Equation (Equation 2)) of 1.31 and a maximum symmetric deviation factor of 2.4. This means the model is on average a factor of 1.31 off-target in either direction.
(2)$$\begin{array}{c} \mathrm{MSDF}=\frac{1}{n}\sum_{i}^{n}\exp\left|\ln\left(\frac{\Delta_{i}\left(\exp\right)}{\Delta_{i}\left(\mathrm{calc}\right)}\right)\right| \end{array}$$

Considering that the splittings span a range of four orders of magnitude, this is a very satisfactory performance for a one-dimensional relaxed DFT-based model. It is somewhat better than the best performing simple model of Ref. [93] (Effective Barrier Height Model, MSDF of 1.39, based on the same computational level and benchmarking set) and has the advantage of requiring no adjustable parameters. However, its disadvantage is that it is more computationally intensive.

Returning to the symmetric limiting case to enable a fair comparison between models, the potential of (+)-α-fenchol is symmetrized by averaging the 180 half-potentials separated at the g−/g+ transition state (Appendix A). For the moment of inertia, the value of the g− conformer is used. The calculated moment of inertia varies only in a narrow corridor of less than ±1% between the stationary points (Appendix A), justifying the approximation of using a constant value that does not depend on the torsional coordinate. The results, given in Table 2, are similar to both the estimations from the simpler models and the experimentally derived torsional splittings, supporting substantial contributions from the tunneling interaction to the latter.

Calculated torsional splittings for the unmodified asymmetric B3LYP potential are 25 cm${}^{-1}hc$ (OH) and 19 cm${}^{-1}hc$ (OD) (Appendix A), larger than the observed ones of 16 and 7 cm${}^{-1}hc$. While the calculated splittings are insensitive to small errors in the barrier height in the order of a few cm${}^{-1}hc$, they are very sensitive to absolute errors of this magnitude for the asymmetry. The conformational energy difference is likely overestimated by the DFT methods (Table 1). Applying DLPNO-CCSD(T)/aug-cc-pVQZ single-point corrections to the six stationary points (minima and transitions states) and linearly scaling the connecting potential parts to match, results in the potential are shown in Figure 10.

The calculated splittings drop to 17.0 (OH) and 5.3 cm${}^{-1}hc$ (OD), in good agreement with the experiment.

In a simple perturbation model, the tunneling splitting in the (hypothetical) symmetric case Δ and the energetic asymmetry between hypothetical localized states δ contribute to the observed total splitting *T* according to Equation (Equation 3) [4,5,34,35]:(3)$$\begin{array}{c} T=\sqrt{\Delta^{2}+\delta^{2}} \end{array}$$

To separate the two contributions, artificial localization is applied to the DLPNO-CCSD(T)//B3LYP-D3(BJ) potential by adding an additional narrow (1${}^{\circ }$) but towering ($10^{6}$ cm${}^{-1}hc$) rectangle barrier at the g−/g+ transition state which suppresses tunneling (Appendix A). This lowers the splitting of the first two torsional states from T=17.0 cm${}^{-1}hc$ to δ= 9.1 cm${}^{-1}hc$, which in turn yields Δ= 14.4 cm${}^{-1}hc$ according to Equation (Equation 3), in good agreement with the results from the other approaches (Table 2). For the deuterated alcohol, artificial localization converts T= 5.3 cm${}^{-1}hc$ into δ= 4.6 cm${}^{-1}hc$, yielding Δ=2.6 cm${}^{-1}hc$, again in decent agreement with the other estimates. With Δ>δ, the protiated alcohol is strongly delocalized, while the deuterated is moderately, in line with integrated probability distribution shares in the g−/g+ potential wells of 69%/31% (OH) and 88%/12% (OD) (Figure 10). To reproduce the same *T* in the absence of tunneling, the g+ well would have to be lifted by 9.5 cm${}^{-1}$ for OH and by 0.8 cm${}^{-1}$ for OD in the same way in which DLPNO refinements were introduced before, without changing the other stationary points. Small deviations of these values from the term T−δ accrue from the different curvatures of the potential wells, which can lead to further interesting effects, as described below.

According to DLPNO-CCSD(T) single point energy calculations, the g+ equilibrium geometry is slightly lower in energy than g−. However, this sequence is inverted when the respective full harmonic zero-point corrections, calculated at any of the tested DFT levels, are added. This same inversion is observed when considering the anharmonic torsional energy in the one-dimensional DLPNO-CCSD(T)//B3LYP-D3(BJ) potential, where the ground-state has a larger probability density in the shallower g− well (Figure 10), in agreement with conclusions from microwave spectroscopy [50]. Different local zero-point energies open up scenarios in which the asymmetry between the localized torsional states is smaller or/and inverted for deuterated α-fenchol. The deuterium atom may be more strongly delocalized than the protium atom or/and might have a higher integrated probability density in a different potential well. Isotopomeric conformational change [111] and isotopomeric polymorphism [112] upon hydroxy group deuteration are not unprecedented, but so far only documented for hydrogen-bonded systems. As explored in the ESI in Appendix A, the window for such an occurrence is small but within the uncertainties of our calculations. The assignment of the main isotopolog based on the substitution geometry and quadrupole constants of the deuterated compound [50] might thus not be entirely reliable.

Unless overcompensated by a higher asymmetry, a decrease in the splitting would be expected upon OH/OD stretch excitation. This is caused by increases of both the adiabatic barrier (higher OH/OD stretching wavenumber at the transition state) and the moment of inertia (elongation of the vibrationally averaged bond) acting as the effective tunneling mass. Because differences in OH stretch fundamental wavenumbers are very similar to differences in B3LYP harmonic wavenumbers, at least for the minima [51], we estimate the torsional potential for the OH stretch excited state by adding the harmonic wavenumbers to the DLPNO-CCSD(T) electronic energies of the stationary points and again linearly scaling the connecting potential parts. This assumes that changes to the shape of the potential, i.e., torsional positions of stationary points, can be neglected. The extension of the OH or OD bond length upon vibrational excitation is estimated by the change of the rotational constants of the hydroxy radical [72,73], as detailed in Appendix A. This simple model captures the available experimental alcohol data for the reduction of splittings upon OH stretch excitation reasonably well for methanol (exp: 9.1 → 6.3 cm${}^{-1}hc$ [97], calc: 9.5 → 7.7 cm${}^{-1}hc$ (Appendix A)) and propargyl alcohol (exp: 22 → 18 cm${}^{-1}hc$ (Section 3.4), calc: 20.4 → 16.9 cm${}^{-1}hc$ (Appendix A)). The same is true for (+)-α-fenchol for both the OH (exp: 16 → 9 cm${}^{-1}hc$, calc: 17.0 → 10.7 cm${}^{-1}hc$) and the OD species (exp: 7 → 3 cm${}^{-1}hc$, calc: 5.3 → 3.0 cm${}^{-1}hc$), with details for the excited states given in Appendix A.

The hypothetical localized g− conformer is calculated to feature a lower energy but higher OH and OD stretching wavenumbers than the g+ conformer. These differences are of similar size, so that the excitation is expected to reduce the asymmetry for both the OH and OD species. Artificial localization (Appendix A) yields δ= 1.6 (OH) and −1.7 cm${}^{-1}hc$ (OD), meaning very close near-degeneracies. Interestingly, for the deuterated alcohol, the sign of the asymmetry is inverted, so that a switch upon OD excitation from predominant g− to g+ character for the torsional ground state is calculated. Equation (Equation 3) yields Δ= 10.5 (OH) and 2.4 cm${}^{-1}hc$ (OD). With Δ≫δ (OH) and Δ>|δ| (OD), both stretch-excited isotopologs feature strongly delocalized torsional wave functions—more so than in the respective ground state, despite the increases of both the barrier height and the moment of inertia from excitation.

As a first approach to evaluate relative transition intensities, torsional Franck–Condon factors (FCF, Equation (Equation 4)) can be analyzed, obtained by factorizing the dependence of the transition moment *M* on the stretching wave function ψ (being evaluated at a fixed torsional dihedral $\tau_{e}$ and depending on the stretching quantum number ν and the stretching coordinate *r*) and the torsional wave function χ (depending on the stretching ν and the torsional quantum number *m* as well as the torsional coordinate τ) (Equation ()). This factorizing relies on the assumption that the dependencies of the dipole moment μ on the stretch and torsional coordinates are separable or the latter is negligible [113,114].
(4)$$\begin{array}{cc} \mathrm{FCF} & ={\langle \chi_{m^{\prime }}^{v^{\prime }}\left(\tau \right)|\chi_{m}^{v}\left(\tau \right)\rangle }^{2} \end{array}$$
(5)$$\begin{array}{cc} M^{2} & ={\langle \psi_{v^{\prime }}^{\tau_{e}}\left(r\right)\chi_{m^{\prime }}^{v^{\prime }}\left(\tau \right)|\mu \left(r,\tau \right)|\psi_{v}^{\tau_{e}}\left(r\right)\chi_{m}^{v}\left(\tau \right)\rangle }^{2}\approx {\langle \psi_{v^{\prime }}^{\tau_{e}}\left(r\right)|\mu \left(r\right)|\psi_{v}^{\tau_{e}}\left(r\right)\rangle }^{2}\;\cdot \;{\langle \chi_{m^{\prime }}^{v^{\prime }}\left(\tau \right)|\chi_{m}^{v}\left(\tau \right)\rangle }^{2} \end{array}$$

The ground and first excited torsional states have preponderantly, but not dominantly, g− and g+ character, respectively, for protiated (+)-α-fenchol in both the OH stretching ground and excited state. The medium-wavenumber transitions $l_{1}\;\leftarrow \;l_{0}$ and $u_{1}\;\leftarrow \;u_{0}$, connecting torsional wave functions with the same character and same number of nodes, have the largest FCFs. FCFs for the satellite $u_{1}\;\leftarrow \;l_{0}$ and $l_{1}\;\leftarrow \;u_{0}$ transitions are about 4% of this size (Appendix A). This is about three times more than for borneol (Appendix A) and about a hundred times more than for isopinocampheol (Appendix A), but clearly less than the observed intensity ratios for (+)-α-fenchol (Figure 4). An error in the calculated asymmetry of the potential is likely not to blame for this deviation, as the agreement cannot be substantially improved by varying the asymmetry in a concerted way for the OH stretch ground and excited state (Appendix A). This underestimation of transition intensities connecting torsional wave function with different number of nodes by the Franck–Condon model is reminiscent of the situation for OH stretching spectra of symmetric hydroperoxides [114,115,116]. It was explained by a neglected component of the transition dipole moment being an antisymmetric function of the torsional coordinate, which needs to be factored into the symmetry selection rules.

In contrast, for the deuterated alcohol, the calculated FCF ratios for both $u_{1}\;\leftarrow \;l_{0}$/$l_{1}\;\leftarrow \;l_{0}$ and $u_{1}\;\leftarrow \;u_{0}$/$l_{1}\;\leftarrow \;u_{0}$ are close to unity (Appendix A), in good agreement with the observed intensity ratios (Figure 6). These values can be understood by the OD torsional wave functions being mostly localized in different wells for the OD stretch ground and excited state, which reduces the partial cancelation between positive and negative contributions of $\chi_{m^{\prime }}^{v^{\prime }}\chi_{m}^{v}$ to the integral. This also lowers the expected impact of neglected sign changes of the transition dipole moment function.

Alternatively, one can also attempt to add the small variation of the non-torsional harmonic zero-point energy of the stationary points to the scaling of the potential. Considering the shortcomings of the harmonic approximation and α-fenchol featuring several low-wavenumber normal modes, in part with multiple modes having some torsional character depending on the conformer, such an adiabatic separation is not necessarily an improvement over the purely electronic potential. This approach leads to similar splittings of T= 17.8 (OH) and 10.0 cm${}^{-1}hc$ (OD) in the ground state as well as 8.5 (OH) and 3.1 cm${}^{-1}hc$ (OD) in the OH and OD stretch excited state, with details given in Appendix A . A qualitative difference is that a conformational inversion upon OD stretch excitation is no longer predicted. This results in larger FCF ratios (more localized intensity) than for the protiated alcohol (Appendix A), contrary to the experimental change upon deuteration.

While both variants of the model reproduce the observed torsional splittings quite well, a more sophisticated future investigation of relative intensities is required to investigate whether the highly unusual OD stretching spectrum is indeed caused by a conformational inversion upon excitation, as suggested by the FCFs. This would probably require both a more accurate potential (for which the relatively large molecular size of α-fenchol (C${}_{10}$H${}_{18}$O) provides a significant challenge) as well as explicit calculation of transition moments. The importance of the latter is further stressed by the result of the next section, where we show that there can be major differences between relative IR and Raman activities in the limiting case of a symmetric alcohol.

### 3.4. Experimental Support from Monomeric Propargyl Alcohol

To also compare the experimental spectra of α-fenchol against those of a fully symmetric alcohol, we investigate propargyl alcohol. While the structural similarity to α-fenchol is limited (Figure 1), we still believe this species to be especially well suited for a number of reasons:The ground-state splitting is well determined by rotational spectroscopy (about 22 cm${}^{-1}hc$) [86,87] and of similar size to the one proposed here for α-fenchol (16 cm${}^{-1}$hc).Propargyl alcohol has a relatively high volatility (normal boiling point 115 ${}^{\circ }C$), which enables to utilize high vapor pressures without heating, which is helpful to retain signal intensity in mild expansions with low carrier gas pressure.The *t* conformer of propargyl alcohol, diastereomeric to the two enantiomeric *g* conformers interacting through tunneling, is calculated to be very high in energy (about 7 kJ mol${}^{-1}$) and therefore does not perturb the spectrum of the *g* conformers.The presence of the π-system leads to both hydroxy groups donating a hydrogen bond each in dimers of propargyl alcohol [117,118]. This shifts their OH stretching fundamentals out of the monomer spectral region, so that the latter can be analyzed without the risk of contamination from signals of dangling hydroxy groups of dimers. This conclusion for propargyl alcohol will be reconfirmed in Section 3.5.

In previous work [51], we investigated Raman jet spectra of propargyl alcohol. Despite only the enantiomeric *g* conformers being energetically relevant, we observed two signals in the monomer OH fundamental region at 3662 and 3658 cm${}^{-1}$ (Figure 11, violet trace). The weaker lower-wavenumber signal rapidly loses relative intensity in colder parts of the expansion more distant from the nozzle. We interpreted this spectral splitting of 4 cm${}^{-1}$ as a lowering of the tunneling splitting by this amount upon OH stretch excitation. The proposed energy level scheme based on these observations is shown in the right part of Figure 11, assigning Raman-active symmetry-conserving transitions ($l_{1}\;\leftarrow \;l_{0}$ and $u_{1}\;\leftarrow \;u_{0}$). The derived OH stretching fundamental wavenumber of a hypothetical localized *g* conformer (3660 cm${}^{-1}$) is in decent agreement with the predicted one by the model for shifting the harmonic PBE0 wavenumber (3657 cm${}^{-1}$) [51]. No Raman activity for conceivable symmetry-changing transitions ($u_{1}\;\leftarrow \;l_{0}$ and $l_{1}\;\leftarrow \;u_{0}$) at 3680 or 3640 cm${}^{-1}$ was observed.

Analogous transitions for hydroperoxides were shown to have non-negligible infrared activity for multi-quanta OH stretching excitation [114,115,116], contrary to the symmetry-selection rules from the simple vibrational Franck–Condon model; for single-quantum OH, stretch excitation even intensities comparable with the symmetry-conserving transitions are predicted [114,116]. Considering the structural similarities between alkyl hydroperoxides and alcohols, we extend here our investigation of propargyl alcohol to FTIR detection. An early grating spectrometer study of propargyl alcohol vapor yielded a double-peak band centered around 3663 cm${}^{-1}$ [119]. A later high-resolution jet study reported a rich amount of transitions in the 3666–3660 cm${}^{-1}$ range but did not explore beyond [120]. Using a lower resolution but a broader spectral range, we indeed observe a band at the expected position of 3680 cm${}^{-1}$ in the cold jet expansion (Figure 11, blue trace), at roughly one fifth of the intensity of the main band. In warmer expansions (green and orange traces), utilizing low carrier gas pressures, the signal is increasingly obstructed by the rovibrational R-branches of the symmetry-conserving transitions and converts to a shoulder in the spectrum of the stationary vapor at ambient temperature (red trace). At the second expected position at 3640 cm${}^{-1}$, a similar shoulder is observed. This second symmetry-changing transition is only weakly visible in the jet spectra, as the originating upper tunneling state is barely populated in a cold expansion and the signal is partially obstructed in the warmer expansions by the P-branches.

Satellites, shifted by +8 and −9 cm${}^{-1}$ from the main band, with similar relative intensities were also observed for propargyl alcohol in an argon matrix but were not discussed [121]. Possible interpretations are that tunneling is quenched by the matrix environment and an unrelated site splitting is present, or that the tunneling splitting is reduced—the latter is observed, but far more drastically, for the propargyl alcohol⋯argon complex [88].

In conclusion, the FTIR jet spectra of propargyl alcohol confirm our previous interpretation of the Raman jet spectra as well as the expectation that symmetry-changing transitions do not follow the simple vibrational Franck–Condon scheme (Appendix A) but carry non-negligible IR activity [114,116]. The FTIR jet spectra of propargyl alcohol and α-fenchol at cold conditions are very similar, both featuring one intense band and a higher-wavenumber less-intense band, separated by the torsional splitting of the excited OH stretching state. In contrast, the Raman jet spectra are quite different. The larger number of observed bands for α-fenchol in part originate from the lower relative energy of the *t* conformer and the dangling hydroxy group of its dimer. Furthermore, additional transitions between the torsional states carry Raman activity, while they do not measurably for propargyl alcohol. This might hint at a symmetry-based Raman selection rule, which is relaxed by the slight asymmetry present in α-fenchol, or it may be based on specific IR and Raman transition moments in each molecular case. In any case, for the more strongly asymmetric borneol and isopinocampheol, no Raman-activity for additional bands could be observed [51]), which is likely due to negligible overlap between the strongly localized torsional wave functions (Appendix A). This highlights the special in-between status of the slightly asymmetric α-fenchol that results in the observed spectral activity for additional transitions.

### 3.5. Computational Results for Propargyl Alcohol Dimers

To assign observed additional downshifted bands in the FTIR and Raman spectra and reconfirm that no dimer signal is expected to perturb the monomer region, we also investigated dimers of propargyl alcohol. Their conformational space was first explored by Mani and Arunan at MP2/6-311+G(3df,2p) level, who reported seven isomers [117], and later expanded to nine isomers by Saini and Viswanathan at M06-2X/6-311++G(d,p) level [118]. These nine isomers were here reoptimized at B3LYP-D3(BJ)/may-cc-pVTZ level and categorized. Building on the nomenclature established for structurally related dimers of 1-indanol [122], benzyl alcohol [15,123] and 1-phenylethanol [15], we first state the conformation of the constituting monomers *gauche* or *trans* in the sequence donor/acceptor, leading to the combinations *gg, gt, tg* and *tt*. This is followed by hom or het, describing whether the signs for the HOC${}_{\alpha }$C${}_{\beta }$ and H${}^{\prime }$O${}^{\prime }$C${}_{\alpha }^{\prime }$C${}_{\beta }^{\prime }$ dihedrals (in a *gg* combination) or for the HOC${}_{\alpha }$C${}_{\beta }$ and the E${}^{\prime }$O${}^{\prime }$C${}_{\alpha }^{\prime }$C${}_{\beta }^{\prime }$ dihedrals (in a *gt* combination) agree or disagree with each other, with E standing for the coordinated oxygen electron pair. Finally, the acceptor types for established hydrogen bonds are stated, in the same sequence as the conformations of their monomer donors, with the four possibilities O${}^{g}$, O${}^{t}$, π and V. O${}^{g}$ and O${}^{t}$ state the qualitative arrangement of the EOC${}_{\alpha }$C${}_{\beta }$ dihedral for the coordinated oxygen electron pair, V stands for a ’void’ acceptor, meaning a dangling hydroxy group. Going through these categorizations, it was noticed that within the nine previously reported isomers several conceivable combinations were not represented, which were subsequently explored. Furthermore, in both previous studies, it was presupposed that energetically relevant isomers have to feature an OH⋯O hydrogen bond. However, for the dimers of 1-indanol, benzyl alcohol and 1-phenylethanol, it has been found that isomers with two OH⋯π hydrogen bonds can be competitive enough to be observed in a jet expansion [15,122,123]. These considerations lead to five new isomers, among them are the third, fourth and sixth most stable overall. The six leading isomers are all composed of two *gauche* monomer conformers and feature two hydrogen bonds. They are the hom and het variants of the O${}^{t}\pi$, O${}^{g}\pi$ and ππ motifs and their structures are given in Figure 12 together with their OH stretching vibrational properties. The homochiral dimerization preference in an O${}^{g}\pi$ motif qualitatively parallels that for benzyl alcohol dimers [15]. The full list of isomers is available in Appendix A.

The calculated vibrational properties promise complementary information from FTIR and Raman jet spectroscopy for the discrimination between the three relevant motifs. In the O${}^{t}\pi$ motif, the two OH stretching oscillators are only weakly coupled and the more strongly downshifted fundamental localized in the OH⋯O hydrogen bond features both higher IR and Raman activity than the less downshifted fundamental for the OH⋯π hydrogen bond. In contrast the two oscillators are strongly coupled in the O${}^{g}\pi$ motif, with the out-of-phase-combination having higher IR activity and the in-phase-combination having higher Raman activity. Finally, in the ππ motif, the OH stretching oscillators are symmetry-equivalent and thus fully coupled with a predicted Davydov splitting of about 10 cm${}^{-1}$; the higher-wavenumber anti-symmetric and the lower-wavenumber symmetric combinations are (almost) exclusively IR or Raman active, respectively.

### 3.6. Vibrational Spectra of Propargyl Alcohol Dimers

The global minimum isomer *gg*-hom-O${}^{g}\pi$ was identified with rotational spectroscopy in a helium expansion by Mani and Arunan [117]. Saini and Viswanathan assigned in addition also the second most stable isomer *gg*-hom-O${}^{t}\pi$ with FTIR spectroscopy in a nitrogen matrix [118]. However, strong overlap between the broad OH stretching signals of the isomers, and also with a signal of the mixed propargyl alcohol-water dimer, required a complex spectral deconvolution which left room for ambiguity. In comparison, the signals observed here in jet expansions are narrower and thus better resolved. The absence of the matrix environment also makes the comparison with theoretical predictions more straightforward. In Figure 13, the experimental spectra (inner traces) are compared with simulations (outer traces).

The four bands at 3581, 3562, 3532 and 3507 cm${}^{-1}$ can be attributed to dimers, in agreement with the number derived from deconvolution of the matrix spectrum. Assuming that the spectral sequence is preserved, matrix shifts relative to the gas phase amount to between −75 and −92 cm${}^{-1}$. The intensity sequence as well as its change upon switch from FTIR to Raman detection are in very good agreement with the simulation, so the two most stable isomers can be assigned with confidence. Downshifts relative to the monomer are overestimated by harmonic B3LYP-D3(BJ) for all four signals, the stronger the higher the OH⋯O character of the fundamental. No signals exclusive to the FTIR or Raman spectrum are observed, indicating that ππ isomers are negligibly populated. This differs from the behavior of benzyl alcohol dimers but is consistent with predicted higher relative energies of these isomers for propargyl alcohol at this level of theory [15]. A band pattern that is even more strongly downshifted appears in the spectral region around 3450 cm${}^{-1}$. It is clearly observed in the FTIR and weakly in the Raman spectrum, and scales more steeply with the propargyl alcohol concentration than the dimer signals (Appendix A). It can therefore be attributed to larger clusters, probably trimers.

In conclusion, the predicted and experimental spectra for propargyl alcohol dimers are in good agreement with each other and substantiate that no relevant population of higher-energetic isomers with dangling hydroxy groups is expected to perturb the monomer region.

### 3.7. Vibrational Spectra of (+)-α-Fenchol Dimers

In the FTIR and Raman jet spectra of protiated (+)-α-fenchol, downshifted bands are observed as well, with the most prominent one at 3499 cm${}^{-1}$ (Figure 14).

This signal is not present in the FTIR spectrum of the vapor at ambient temperature and gains relative intensity with a higher expansion pressure and a detection farther from the nozzle. It is assigned to the OH stretch of the donor hydroxy group in the main isomer of the dimer of (+)-α-fenchol. Additional weak signals in the same spectral region indicate further minor isomers. The 3499 cm${}^{-1}$ donor signal notably broadens and upshifts when probed very close to the nozzle, which also applies to a lesser extent to the acceptor signal at 3632 cm${}^{-1}$. This might be caused by thermal excitation of low-wavenumber intermolecular vibrations which weakens the hydrogen bond.

In Figure 14, the FTIR and Raman jet spectra of protiated and deuterated (+)-α-fenchol are compared by aligning the wavenumber axes according to $0.73778\;\cdot \;\tilde{\nu }\left(\mathrm{OH}\right)=\tilde{\nu }\left(\mathrm{OD}\right)$, as suggested by the model of Figure 7. This highlights the close resemblance in the monomer and dimer acceptor spectral region, barring the different tunneling splittings and impurities. For the deuterated dimer, signals are observed at 2680 and 2585 cm${}^{-1}$ for the acceptor and donor, respectively. The position of the latter is underestimated by 4 cm${}^{-1}$ by extrapolation of the model for a fixed fundamental ratio for dangling hydroxy groups. About half of this discrepancy might be explained by a substantially lower harmonic wavenumber, which impacts the isotope effect according to Equation (Equation 1). An increase of diagonal anharmonicity upon hydrogen-bonding, as concluded from overtone data for other alcohol dimers [78], might also contribute.

### 3.8. Computational Results for (+)-α-Fenchol Dimers

For a hydrogen-bonded dimer of (+)-α-fenchol, three main conformational degrees of freedom need to to be considered. First, both the donor and acceptor can adopt any of the three intramolecular conformations g−, g+ or *t*, resulting in nine combinations. Second, either of the two diastereomeric oxygen electron lone pairs can be coordinated. Third, multiple isomers may exist regarding the torsion about the hydrogen bond, as it can be described by the C${}_{\alpha }$OO${}^{\prime }$C${}_{\alpha }^{\prime }$ dihedral. Systematic exploration of this conformational space at the B3LYP-D3(BJ)/def2-TZVP level led to 51 isomers spanning a range of 11 kJ mol${}^{-1}$. The six most stable up to 3.1 kJ mol${}^{-1}$ were reoptimized with the larger basis set may-cc-pVTZ at the B3LYP-D3(BJ) and PBE0-D3(BJ) levels, listed in Appendix A.

The structure of the most stable conformer that lies 1.5 (B3LYP) or 1.4 kJ mol${}^{-1}$ (PBE0) lower in zero-point corrected energy than the second is shown in Figure 15.

It constitutes a g− conformer as the donor and a *t* conformer as the acceptor for the hydrogen bond. This preference for the g− conformer in the donor role has been previously observed for mixed dimers of (+)-α-fenchol with (+)- and (−)-α-pinene [74], along with a switch to the *t* conformer when in the acceptor role for the complex of α-fenchol with water [124]. The harmonic OH/OD stretching wavenumber of the dangling hydroxy group is predicted to be downshifted by 4/3 (B3LYP) or 7/5 cm${}^{-1}$ (PBE0) from the isolated *t* conformer, which is in qualitative agreement with the observed downshifts for the fundamental transitions of 12/9 cm${}^{-1}$. The donor and acceptor modes are predicted to be decoupled enough that OD stretching signals from single- and double-deuterated dimers are expected to coincide within our spectral resolution. Within our signal-to-noise ratio, the dimer acceptor band is observed in the Raman but not in the FTIR jet spectrum. This is consistent with its Raman activity being predicted to be about four times smaller than for the donor, whereas its IR activity is predicted to be about 25 times lower.

The assignment to this structure is plausible but not rigorous due to the fact that calculated OH stretching properties for dimers are less reliable than those for monomers. Microwave confirmation of this assignment would be welcome. This might also be interesting in the light of one particular detail: if one disregards the two hydroxy hydrogen atoms, then the system becomes almost $C_{2}$-symmetric (Figure 15). We speculate that these two hydrogen atoms might be yet again delocalized in this heavy-atom frame, this time within a geared internal rotation which interconverts donor and acceptor, similar to intermolecularly hydrogen-bonded dimers of hydrogen halides [22,35,79] and intramolecularly hydrogen-bonded diols [125,126,127,128], potentially detectable as a splitting with high-resolution spectroscopy.

## 4. Conclusions and Outlook

In this article, we investigated the hypothesis that tunneling between two accidentally near-resonant torsional conformers of the monoterpenol α-fenchol explains its anomalous FTIR and Raman jet spectra in the OH and OD stretching region. Tunneling-mediated relaxation of torsionally-excited states helps understand why a second low-lying conformer (in the order of 0.1 kJ mol${}^{-1}$) has not been observed in microwave jet spectroscopy [49,50]. By replacing neon with helium as the carrier gas at a lower pressure and detection close to the nozzle, we were able to observe vibrational transitions originating from a second low-lying torsional state, which may be interpreted as the elusive second conformer. However, on the balance of the evidence presented here, we suggest that these torsional states are in fact delocalized across both potential wells. This partial delocalization is expected to transfer spectral activity to additional transitions, compared to both more strongly asymmetric and fully symmetric species, as validated with the examples of borneol, isopinocampheol, and propargyl alcohol.

Observed spectral splittings in OH stretching bands could be translated into torsional splittings, determined for the main isotopolog of α-fenchol as 16(1) cm${}^{-1}$ and 9(1) cm${}^{-1}$ for the OH stretch ground and excited state, respectively. As the tunneling interaction largely contributes to these splittings, they are characteristically [93] reduced to 7(3) and 3(1) cm${}^{-1}$ upon increasing the tunneling mass by hydroxy deuteration. Despite being expected to be less delocalized, the deuterated alcohol shows a more even distribution of Raman activity across the split bands. Based on calculated torsional Franck–Condon factors, this might be explained by a switch of the conformational preference upon OD stretch excitation.

Due to known shortcomings of the Franck–Condon model [114,115,116] and the sensitivity on the details of the potential, the latter hypothesis, however, requires further investigation, especially explicit theoretical modeling of relative IR and Raman intensities. Further experimental insights might come from microwave characterization of the second and perhaps also the third torsional state populated in a milder jet expansion or at ambient temperature, a direct measurement of the mm-wave or THz torsional transitions expected at 16(1) (OH) and 7(1) cm${}^{-1}$ (OD), as well as analysis of OH and OD stretching overtone spectra.

Modification of the carbon skeleton, converting α-fenchol into borneol or isopinocampheol, was found to lead to localization and simpler spectra. The rather large molecular structure of α-fenchol offers a multitude of additional positions for chemical or isotopic modification to fine-tune the torsional splittings and hydrogen (de-)localization, challenging theory for the calculation of conformational/torsional differences on a 1 cm${}^{-1}hc$ ($\hat{\approx }$0.01 kJ mol${}^{-1}$) scale. For propargyl alcohol, the deuteration of one of the two methylene hydrogen atoms is calculated to introduce an isotopic zero-point asymmetry smaller than the tunneling splitting of the symmetric main isotopolog. We therefore expect this isotopolog to feature a complex OH stretching Raman jet spectrum similar to the one of α-fenchol.

In conclusion, the presented analysis highlights the importance of considering nuclear quantum effects for comprehension of the structure and dynamics of matter, even for cases such as asymmetric biomolecules with their (seemingly) well-defined geometries.

## Figures and Tables

**Figure 1 molecules-27-00101-f001:**
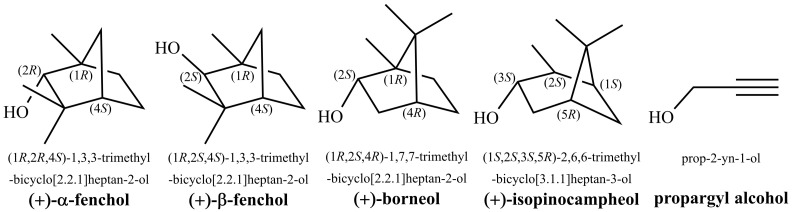
Structural formulas, systematic names and used trivial names of alcohols discussed in this study.

**Figure 2 molecules-27-00101-f002:**
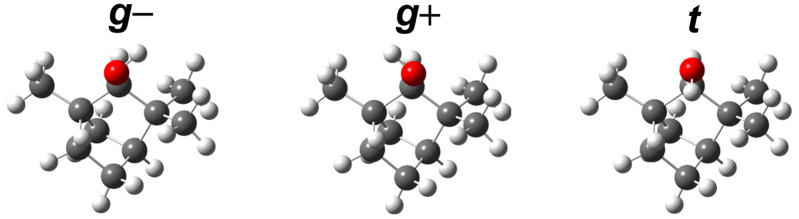
Optimized torsional conformers of (+)-α-fenchol at B3LYP-D3(BJ)/may-cc-pVTZ level. They are labeled according to the qualitative arrangement of the HOC${}_{\alpha }$H dihedral: gauche (*g*) ≈−60 or +60, or trans (*t*) ≈±180.

**Figure 3 molecules-27-00101-f003:**
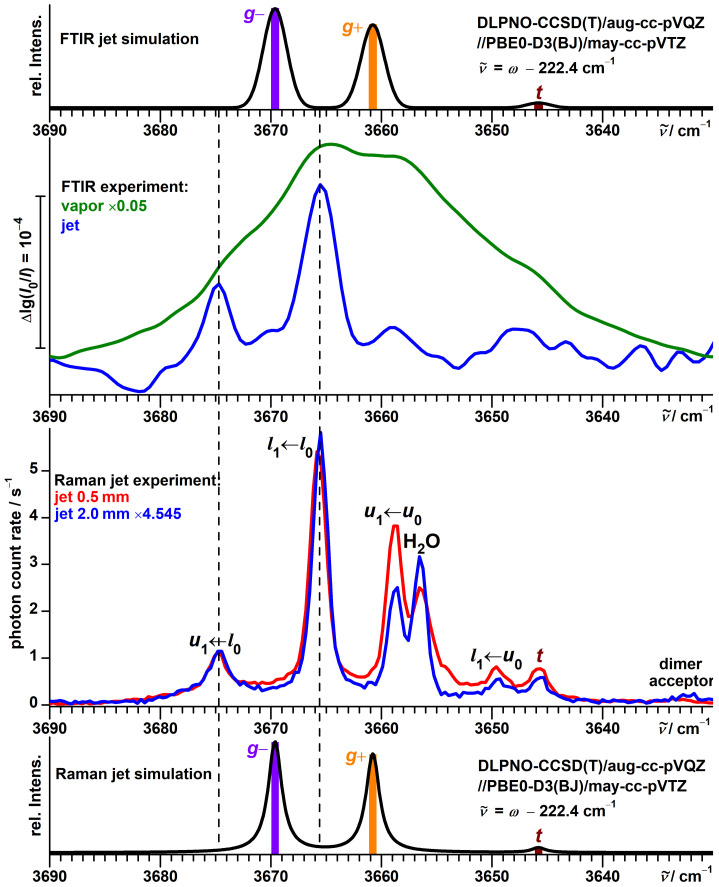
Top half: Comparison between simulated and experimental FTIR jet spectra of α-fenchol. In addition, an FTIR spectrum of the vapor at ambient temperature is shown. Bottom half: Comparison between simulated and experimental Raman jet spectra of α-fenchol at two different detection distances from the nozzle. For the simulations, a Boltzmann distribution of localized conformers is assumed at a conformational temperature of 100 K. Harmonic OH stretching wavenumbers are uniformly shifted according to the model of Ref. [51].

**Figure 4 molecules-27-00101-f004:**
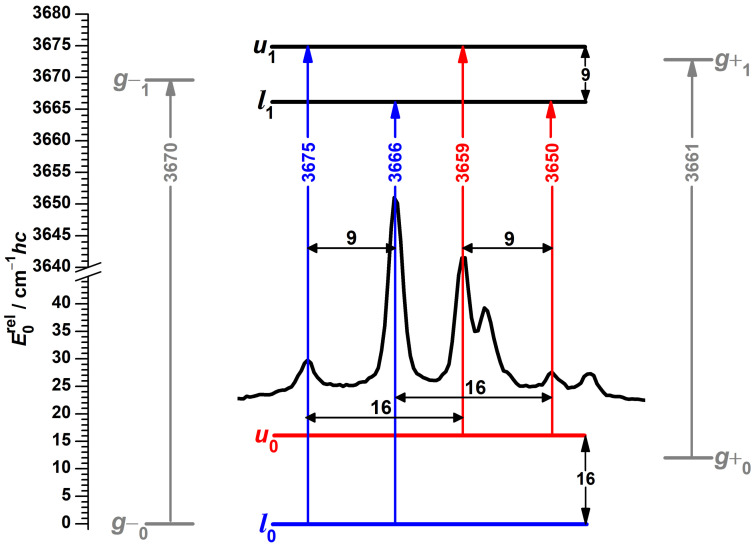
Raman spectral pattern in the OH stretching spectrum of α-fenchol with repeating separations (black horizontal double-headed arrows) translated into torsional splittings (black vertical double-headed arrows). The ground state $l_{0}$ and transitions originating from it are colored blue, for the first excited torsional state $u_{0}$ likewise red, for hypothetical localized g− and g+ states gray, the latter based on DLPNO-CCSD(T)/aug-cc-pVQZ//PBE0-D3(BJ)/may-cc-pVTZ calculations with shifted harmonic OH stretching wavenumbers [51]. *l* stands for lower and *u* for upper torsional state, the index 0 or 1 indicates the OH stretch quantum number.

**Figure 5 molecules-27-00101-f005:**
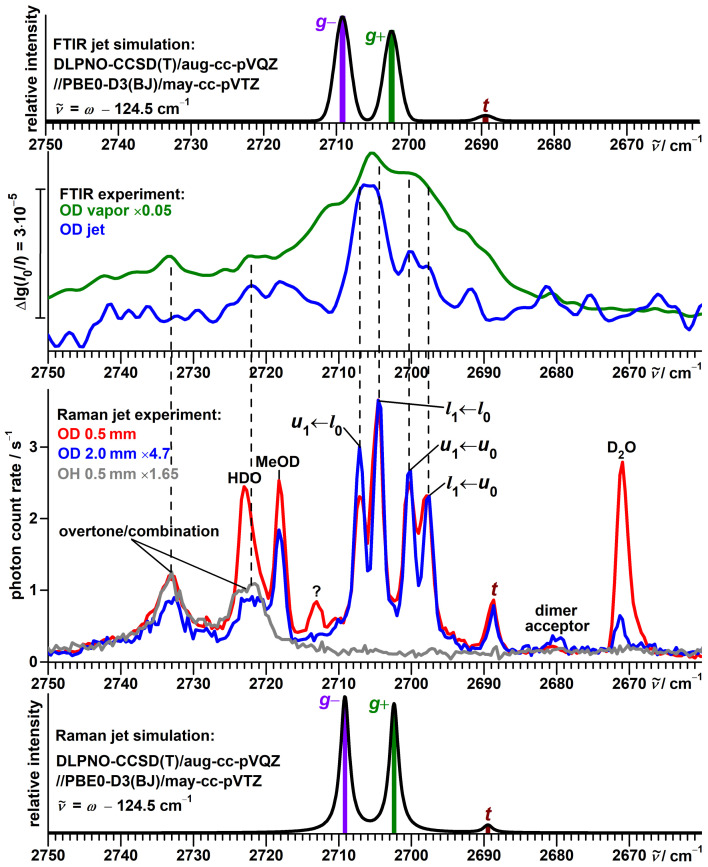
Top half: Comparison between simulated and experimental FTIR jet spectra of α-fenchol-OD. In addition, an FTIR spectrum of the vapor at ambient temperature is shown. Bottom half: Comparison between simulated and experimental Raman jet spectra of α-fenchol-OD at two different detection distances from the nozzle in the OD stretching region. In addition, a spectrum of α-fenchol-OH in this spectral region is shown. For the simulations, a Boltzmann distribution of localized conformers is assumed at a conformational temperature of 100 K. Harmonic OD stretching wavenumbers are uniformly shifted according to the model of Ref. [51].

**Figure 6 molecules-27-00101-f006:**
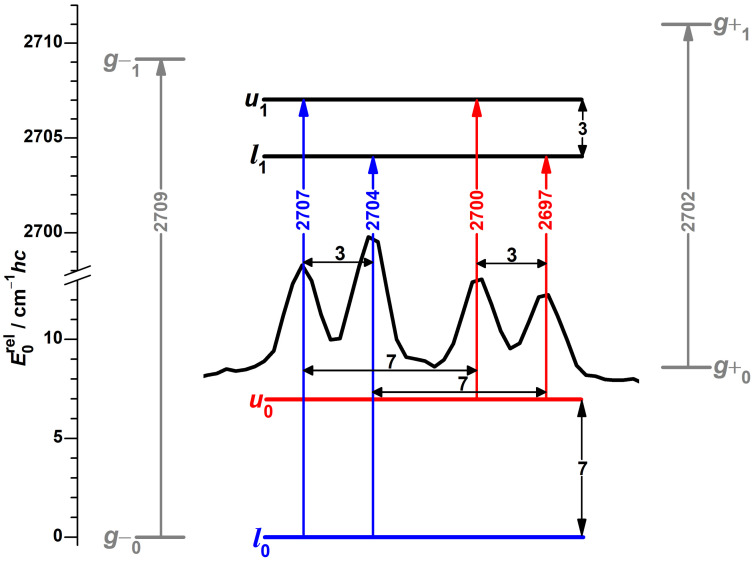
Raman spectral pattern in the OD stretching spectrum of α-fenchol with repeating separations (black horizontal double-headed arrows) translated into torsional splittings (black vertical double-headed arrows). The ground state $l_{0}$ and transitions originating from it are colored blue, for the first excited torsional state $u_{0}$ likewise red, for hypothetical localized g− and g+ states gray, the latter based on DLPNO-CCSD(T)/aug-cc-pVQZ//PBE0-D3(BJ)/may-cc-pVTZ calculations with shifted harmonic OD stretching wavenumbers [51]. *l* stands for lower and *u* for torsional upper state, the index 0 or 1 indicates the OD stretch quantum number.

**Figure 7 molecules-27-00101-f007:**
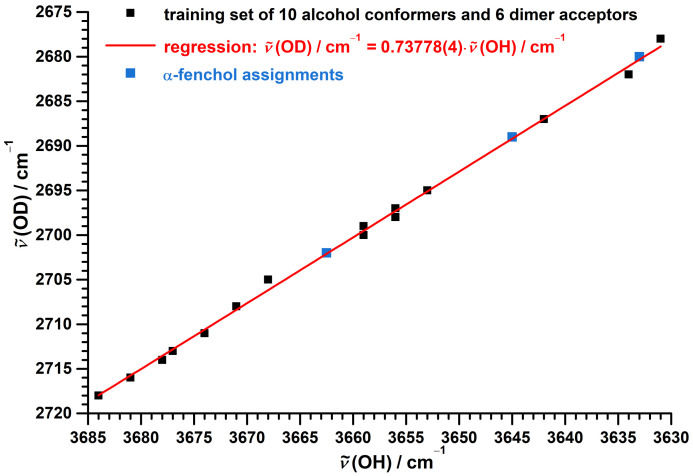
Correlation between experimental jet OD and OH stretching fundamental wavenumbers of alcohol monomers and alcohol dimer acceptors. Used data [75,76,77,92] for the training set are listed in Appendix A. The linear regression for the training set uses a fixed intercept of zero.

**Figure 8 molecules-27-00101-f008:**
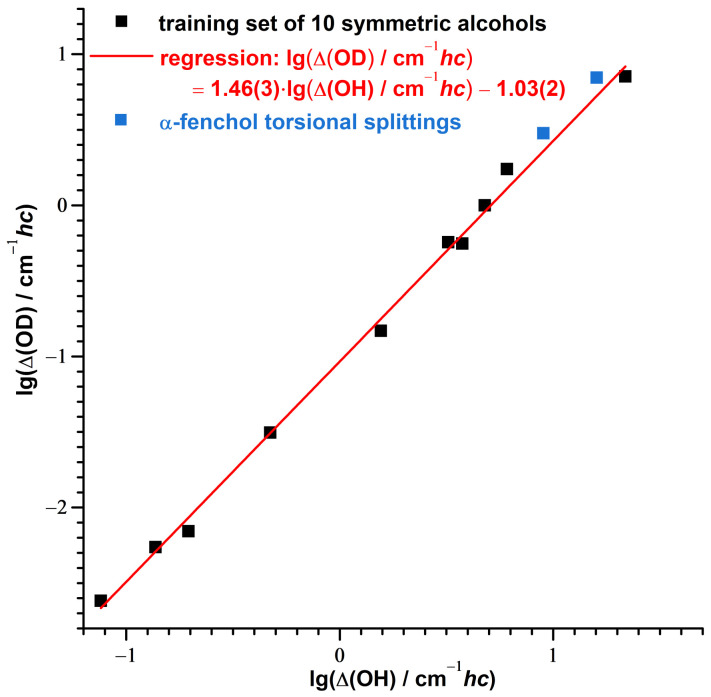
Correlation between the logarithmized experimental tunneling splittings $\mathrm{lg}(/{\mathrm{cm}}^{-1}hc)$ of deuterated and protiated symmetric alcohols (Direct Correlation Model of Ref. [93], details given there) compared with the torsional splittings assigned for the ground and the OH/OD stretch excited state of α-fenchol.

**Figure 9 molecules-27-00101-f009:**
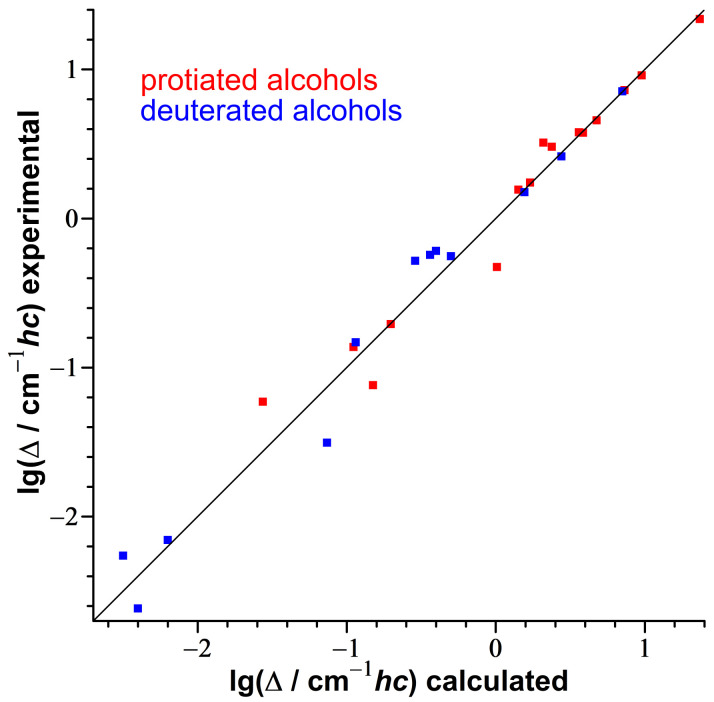
Correlation between logarithmized experimental tunneling splittings [20,87,94,95,96,97,98,99,100,101,102,103,104,105,106,107,108,109,110] and calculated ones with the 1D torsion code based on electronic B3LYP-D3(BJ)/may-cc-pVTZ potentials for 27 isotopologs of 16 symmetric alcohols. The diagonal line represents perfect agreement. Data points are given in Appendix A, diagrams with torsional potentials, energy levels, probability distributions and moment of inertia for all species are available in Appendix A.

**Figure 10 molecules-27-00101-f010:**
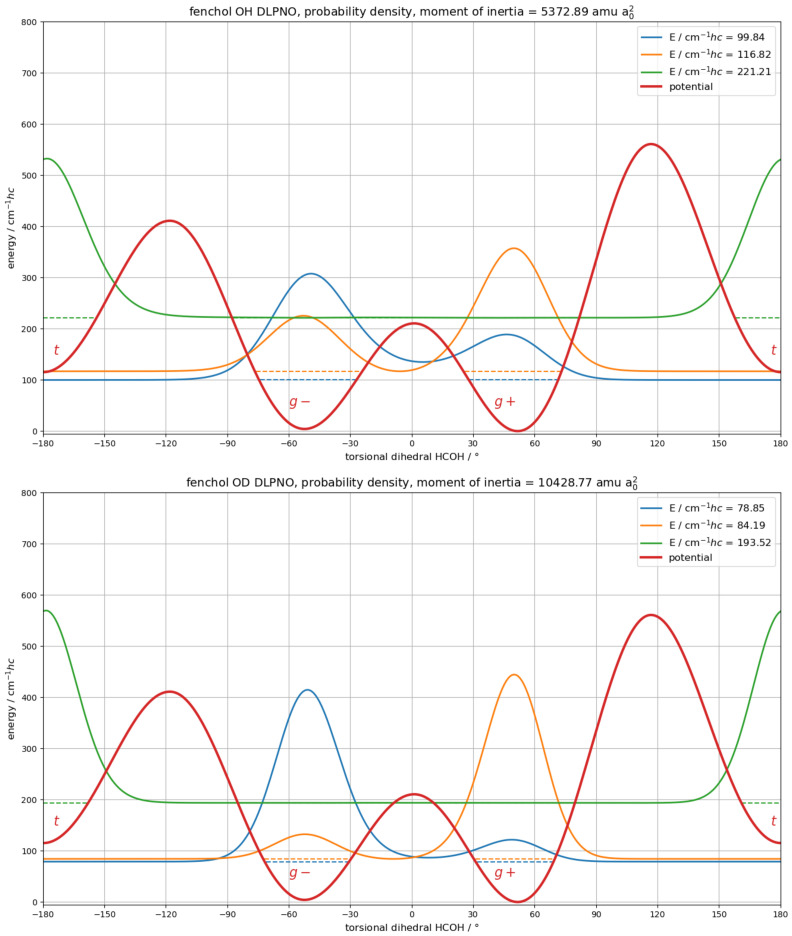
Electronic torsional potential (red trace) of (+)-α-fenchol calculated at B3LYP-D3(BJ)/may-cc-pVTZ level and scaled to DLPNO-CCSD(T)/aug-cc-pVQZ single-point corrections for the six stationary points. A constant moment of inertia, as calculated for the g− minimum geometry, was used. For the protiated (top) and the deuterated alcohol (bottom), the two lowest torsional states (blue and orange horizontal dashed traces) are separated by 17.0 (OH) and 5.3 cm${}^{-1}hc$ (OD) and their probability densities (blue and orange solid traces) are strongly (OH: 69%/31%) to moderately (OD: 88%/12%) delocalized between the g−/g+ potential wells. In contrast, for the third torsional state, the probability density (green solid trace) is very strongly localized (>99%) in the *t* potential well.

**Figure 11 molecules-27-00101-f011:**
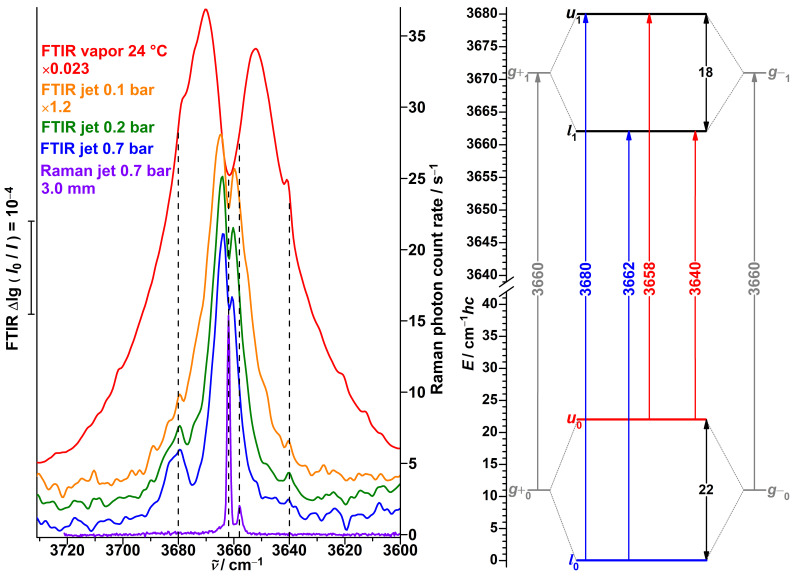
Left: FTIR jet and vapor spectra as well a Raman spectrum of propargyl alcohol in the OH stretching monomer spectral region. Assigned positions of band centers are indicated with dashed lines. Right: Energy level scheme with the ground state and transitions originating from it in blue, likewise for the upper tunneling state in red, as well as for hypothetical localized states in gray.

**Figure 12 molecules-27-00101-f012:**
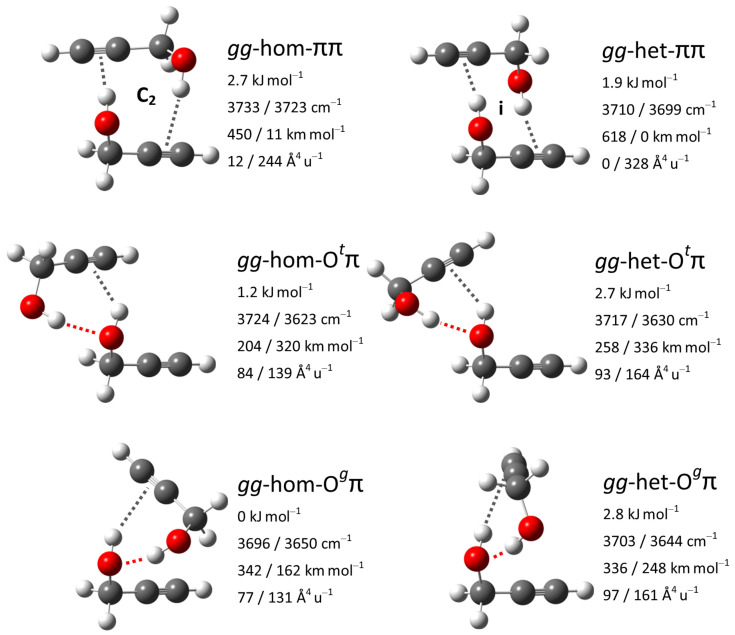
Leading propargyl alcohol dimer structures with harmonically zero-point-corrected relative energies, unscaled harmonic OH stretching wavenumbers as well as associated IR intensities and Raman activities at B3LYP-D3(BJ)/may-cc-pVTZ level. Indicated, where present, are symmetry elements as well as OH⋯O hydrogen bonds by red and OH⋯π by gray dotted lines.

**Figure 13 molecules-27-00101-f013:**
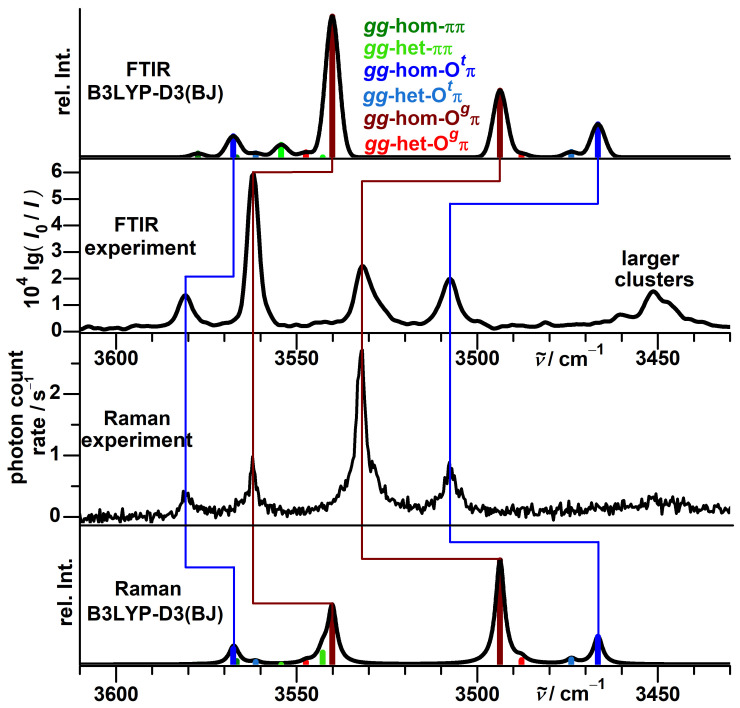
Comparison between experimental (inner) and simulated (outer) FTIR and Raman jet spectra for propargyl alcohol dimers. The simulations assume a conformational temperature of 100 K and identical vibrational and rotational partition functions except for the rotational symmetry number. In addition, the achirality (non-degeneracy) of *gg*-het-ππ is considered in the statistical weight. Empirically applied are a Gaussian FWHM of 4.5 cm${}^{-1}$ for the FTIR simulation and a Lorentzian FWHM of 3 cm${}^{-1}$ for the Raman simulation. Harmonic OH stretching wavenumbers are shifted by −156 cm${}^{-1}$ as needed to match the *g* monomer spectral center of gravity.

**Figure 14 molecules-27-00101-f014:**
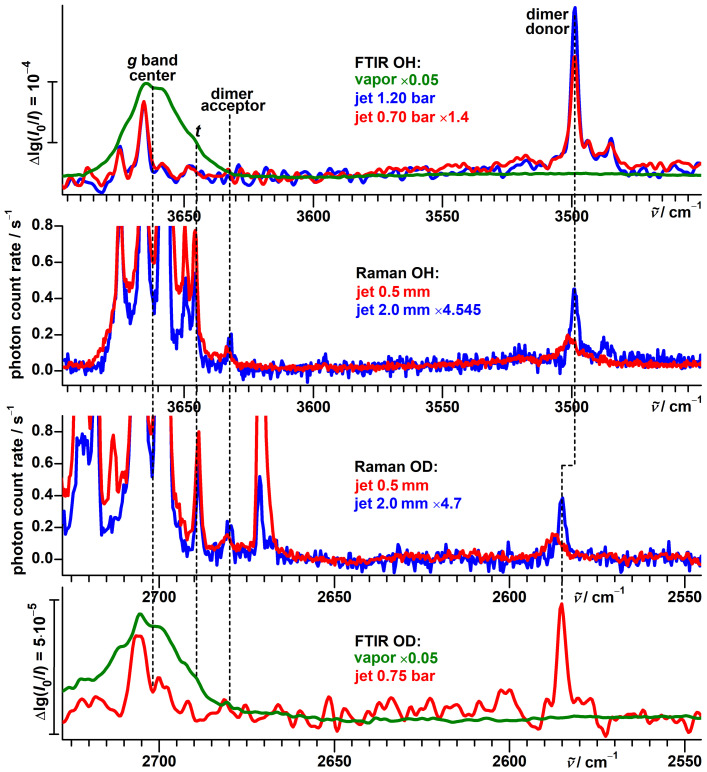
Comparison between FTIR and Raman spectra of protiated (top traces) and deuterated (bottom traces) (+)-α-fenchol. The wavenumber axes for the OH and OD stretching regions are aligned according to $0.73778\;\cdot \;\tilde{\nu }\left(\mathrm{OH}\right)=\tilde{\nu }\left(\mathrm{OD}\right)$.

**Figure 15 molecules-27-00101-f015:**
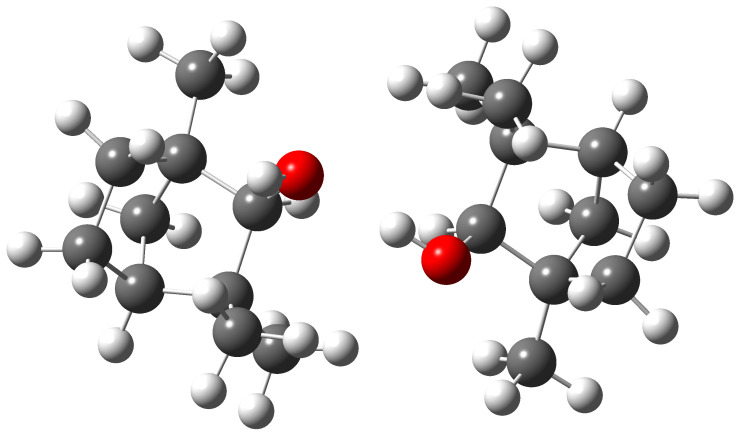
Most stable dimer of (+)-α-fenchol at B3LYP/may-cc-pVTZ level viewed along the approximate $C_{2}$ symmetry axis.

**Table 1 molecules-27-00101-t001:** Differences between α-fenchol conformers relative to g− for harmonically zero-point corrected energies and harmonic OH stretching wavenumbers at different computational levels. For values in parentheses, the electronic contributions to the energy difference were replaced with DLPNO-CCSD(T)/aug-cc-pVQZ single-point values calculated on the respective geometries. A systematic underestimation of OH stretching wavenumber differences between *g* and *t* conformers of alcohols at MP2 level was reported before [51].

Method	$E_{0}\left(g+\right)$ $/\mathrm{kJ}\;\mathrm{mol}{}^{-1}$	$\omega_{\mathrm{OH}}\left(g+\right)$ $/{\mathrm{cm}}^{-1}$	$E_{0}\left(t\right)$ $/\mathrm{kJ}\;{\mathrm{mol}}^{-1}$	$\omega_{\mathrm{OH}}\left(t\right)$ $/{\mathrm{cm}}^{-1}$
MP2/6-311++G(d,p)	0.04 (−0.04)	−8	2.71 (1.61)	−3
PBE0-D3(BJ)/may-cc-pVTZ	0.28 (0.14)	−9	0.37 (1.43)	−24
B3LYP-D3(BJ)/may-cc-pVTZ	0.33 (0.12)	−8	0.81 (1.52)	−17
B2PLYP-D3(BJ)/may-cc-pVTZ	0.22 (0.07)	−8	1.51 (1.54)	−12

**Table 2 molecules-27-00101-t002:** Predicted tunneling contributions Δ to the ground-state torsional splitting of α-fenchol-OH and -OD according to different simple models [93] and the torsion 1D code based on B3LYP-D3(BJ)/may-cc-pVTZ calculations, in part with DLPNO-CCSD(T)/aug-cc-pVQZ corrections. For the simple models, barrier heights and torsional wavenumbers were averaged between g− and g+ (used values given in Appendix A), for the torsion 1D code the torsional potential was either symmetrized by averaging both half-potentials or an additional barrier at the g−/g+ transition state was added to compare with the localized scenario.

Computational Level	Model	$\Delta_{\mathrm{OH}}$/ cm${}^{-1}\mathrm{hc}$	$\Delta_{\mathrm{OD}}$/ cm${}^{-1}\mathrm{hc}$
B3LYP	Eckart Barrier [93]	16.2	3.6
B3LYP	Barrier Height [93]	11.1	5.3
B3LYP	Restricted Barrier Height [93]	15.4	5.1
B3LYP	Effective Barrier Height [93]	17.8	7.3
B3LYP	torsion 1D code symmetrization	15.4	3.4
DLPNO-CCSD(T)//B3LYP	torsion 1D code symmetrization	15.9	3.5
DLPNO-CCSD(T)//B3LYP	torsion 1D code localization	14.4	2.6

## Data Availability

Relevant data are contained within the article and the Appendix A. In addition, all spectra and leading dimer structures are provided in digital format (.dpt, .xyz) at https://doi.org/10.25625/WMUYHZ (accessed on 18 December 2021). The 1D torsion code is available on github, at https://github.com/dlc62/Torsion1D (accessed on 18 December 2021). Spectra of α-fenchol-OH and the tunneling hypothesis were discussed before in the German language as part of the doctoral thesis of Robert Medel [129]. Supporting spectra of the deuterated compound and explicit torsional modeling are presented here for the first time.

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
