# Peer review of "Hydrogen Delocalization in an Asymmetric Biomolecule: The Curious Case of Alpha-Fenchol"

_molecules, 2021, doi:10.3390/molecules27010101_

Round 1

Reviewer 1 Report

General comments: The article “Hydrogen Delocalization in an Asymmetric Biomolecule: The Curious Case of alpha-Fenchol” investigates the FTIR and Raman jet spectra of monoterpenol -fenchol using a hypothesis of tunneling between two accidentally near-resonant torsional conformers. The authors have done Raman jet spectroscopy before for other conformers and now continue the study on monoterpenol -fenchol. The article is well-written and deserves publication in the journal Molecules after minor revisions. I recommend the following:

  1. Stating brief information, importance of monoterpenol α-fenchol conformers in the abstract.
  2. Line 96-99: The authors need to be more specific about what they mean with their metaphors.
  3. In terms of purity (lines 207-213), why the authors did not attempt to purify the chemicals?

4.Line 228, Is this interpretation made before in the literature?

Reviewer 2 Report

The present manuscript describes the study of a non-rigid fenchol molecule and several related species, as well as the deuterated isotopomers, with the aid of vibrational spectroscopy and quantum chemical calculations. Both experimental and computational parts look very solid, e.g., the authors used the modern DLPNO-CCSD(T) modifications to refine the DFT single-point energies. The paper is very well written and the authors are clearly experts in their field. All procedures are well described and scientifically sound. The conclusions are well stated, significant and fully supported by the data that they assembled. There really is nothing that I have to criticize and hence I support publication in the present form. 
One very minor comment - The abbreviation list is probably not necessary. 

Reviewer 3 Report

Inspired by the missing observation of the g+ conformer of alpha-fenchol in the neon expansion with microwave spectroscopy, the authors present a detailed study on the hydroxy torsional states in several kinds of alcohols by means of vibrational jet spectroscopy (Raman and FTIR) and quantum mechanical calculations. In order to explain observed data, the authors propose a model based on the delocalization of the hydroxy hydrogen atom through quantum tunnelling between the two non-equivalent (g+ and g-) but accidentally near-degenerate conformers separated by a low and narrow barrier.

While I am unable to assert that the model is right, I can definitely say that the data analysis is robust and the model is reasonable. I think this work will stimulate the scientific community to experiment to see if the proposed model is correct.

The paper is very well written and the results are presented in a very clear manner, I suggest only a few changes:

pag. 2 line 71 and 79: dipole moment → electric dipole moment

pag. 16 eq. 4: add a vertical bar between the bra and the ket

I believe that these data will be of great interest to the audience of Molecules and I recommend the paper for publication as it is.

Reviewer 4 Report

This paper presents interesting and valuable results. It is well written and organized. The discussion is clear, deep, and very detailed. Conclusions are supported by experimental findings. This is a rather long paper that includes spacious supplementary materials. The experimental results are discussed in the context of theoretical calculations at modern levels of theory. The number of references is also big and impressive. The figures are of very good quality. This article can be treated as a mini-review. It is a really nice work.  In my opinion, in can be published in Molecules journal.

I have found only small errors:

Page 9 line 272. It seems that the figure is not 6 ( Raman) but Figure 5 (FT-IR)

Page 20 line 650 there should be Figure S40 instead of Figure 40

Page 24 line 765 the word “cases” is doublet.

Figure S40 caption. There should be: ‘It can be seen that...’ not ‘It can be seen that...’
